# Prevalence of stillbirth and associated factors among deliveries attended in health facilities in Southern Ethiopia

Jegnaw Wolde[1], Dereje Haile[2]*, Kebreab Paulos[3], Mihiretu Alemayehu[2], Asrat Chernet Adeko[2], Asaminew Ayza[1]

1 Wolaita Zone Health Department, Wolaita Sodo, Ethiopia, 2 Department of Public Health, College of Health and Medical Sciences, Wolaita Sodo University, Sodo, Ethiopia, 3 Department of Midwifery, College of Health and Medical Sciences, Wolaita Sodo University, Sodo, Ethiopia

* derehaile2010@gmail.com

## Abstract

### Background

Stillbirth is an unfavorable outcome of pregnancy, which is still prevalent in many countries despite remarkable efforts made to improve the care of pregnant women. While producing estimates consistent with other national reports, all are hindered by limited data and important causes of death are likely to be missed. However; there is a scarcity of data on stillbirth in Ethiopia particularly in the Wolaita zone.

### Objective

To assess the prevalence and associated factors of stillbirth among women giving birth at public hospitals in the Wolaita zone, southern Ethiopia.

### Methods

A facility-based cross-sectional study was conducted in public hospitals in the Wolaita zone. A stratified sampling technique was used to select 737 women. A pre-tested interviewer-administered questionnaire was used for data collection. Data were entered using Epidata version 3.1 and analyzed using SPSS version 20. Bivariate and multiple logistic regression analysis were used and the crude and adjusted odds ratios at a 95% confidence interval with P-value <0.05 were considered to declare the result as statistically significant.

### Result

This study reported an 8.7% [95% CI: 6.5–10.8] prevalence of stillbirth. Women who lived in rural areas, had pregnancy and labor complications, a high number of pregnancies, a prior history of stillbirth, and a complicated delivery were associated with stillbirth. When compared to urban residents, being a rural resident increased the risk of stillbirth by 2.57 fold [adjusted OR = 2.57, 95% CI: 1.23, 5.40]. When compared to their counterparts, women who experienced complications during pregnancy and labor increased 6.23 fold [AOR = 6.23, 95% CI: 2.67–14.58], having a previous history of stillbirth increased 6.89 fold [AOR =

**Data Availability Statement:** All relevant data are within the manuscript and its Supporting Information files.

**Funding:** The author(s) received no specific funding for this work.

**Competing interests:** The authors have declared that no competing interests exist.

**Abbreviations:** JHPIEGO, John Hopkins Program for International Education in Gynecology Obstetrics; ANC, Antenatal Care; HEWS, Health Extension Workers; MCH, Maternal and Child Health; TBA, Traditional Birth Attendant; MMR, Maternal Mortality Ratio; FMOH, Federal Ministry of Health; SDG, Sustainable Development Goal; SNNPR, Southern Nation Nationality of People and Republic.

6.89, 95% CI: 2.57–13.57], and the type of delivery increased 7.13 fold the risk of stillbirth [AOR = 7.13, 95% CI: 2.71–18.73].

## Conclusion and recommendation

The prevalence of stillbirth among women who gave birth in public hospitals in the Wolaita zone was found to be high compared to national and regional figures. Therefore, the federal and regional governments should strengthen inter-sectoral collaboration with health facilities to promote the maternal and health care services utilization. The zonal health department and other concerned bodies should focus on the implementation of the strategies and policies that address and reduce the causes of stillbirth.

## Introduction

According to the world health organization's (WHO) definition, Stillbirth is a baby born with no signs of life at or after 28 weeks of gestation. The burden of stillbirth has remained very high in Sub-Saharan African countries; Ethiopia ranked fifth among the top ten countries with the highest stillbirth rates in the world [1]. Similarly, Ethiopia is the third high-burdened country in east Africa next to Djibouti and Somalia [2]. Efforts to reduce deliveries ending up by stillbirths have not shown much progress in Ethiopia [3].

Stillbirth is an unfavorable outcome of pregnancy, which is still prevalent in many countries despite remarkable efforts to improve the care of pregnant women. It is a neglected tragedy with an estimated 2.6 million deaths per year worldwide and seven thousand women experience a stillbirth a day globally [2]. All most all (99%) occur in low and middle-income countries and 60% occur in rural settings. Several studies in low and middle-income countries [4–6] suggested various determinants for stillbirths. These mainly include lack of accessible obstetric care and inadequate health care service. Furthermore, maternal infections and complications including antepartum vaginal bleeding, pregnancy-related hypertension, failure to give good care during labor and delivery, fetal growth restriction and congenital anomalies and socio-demographic conditions (age, parity, religion, residence, and health service access) are the most important risk factors [5, 6]. In low and middle-income countries, by far the largest source of data on stillbirths comes from population-based household surveys [7]. Although studies have identified several causes of stillbirth, there exists little evidence on the causes of stillbirth in Ethiopia.

The 2016 Ethiopian demographic and health survey (EDHS) showed that the stillbirth rate of Ethiopia was 11.8 per 1000 pregnancies [8]. There is a problem with appropriate data on stillbirth in developing countries such as Ethiopia. While producing estimates consistent with other national reports, all are hindered by limited data and important causes of death are likely to be missed. The data generally available in low and middle-income countries to inform the cause of death often includes only a limited obstetric and/or reproductive history [4]. A recent study from Ethiopia showed low stillbirth rate among women getting prenatal care. Nevertheless, delivery in a health institution did not decrease the chance of having a stillbirth compared with home deliveries [3]. Evidence reported that obstetric emergencies, including antepartum hemorrhage and maternal hypertensive disorders (preeclampsia and eclampsia), are important contributors of stillbirth [9]. Therefore, it is better to improve obstetric and reproductive conditions and other factors such as health service access, behavioral factors, and socio-demographic conditions.

There is limited evidence on factors associated with stillbirth occurring in health facility settings and most of the existing evidences are retrospective surveys [6, 10] which are associated with recall bias. One study by Dejene Tilahun et al. shows the incidence and determinants of stillbirth. This study doesn't appraise the appropriateness of health service access and doesn't provide pieces of evidence of behavioral factors. Besides most of the data are not recorded and sources of data very difficult to obtain. As a result, there is no evidence of the magnitude and causes of stillbirth among women who gave birth in public and private hospitals in the study area. Stillbirths are increasingly being counted at a local level; however, the absence of global goals and reporting mechanisms continues to restrict their visibility, especially in the countries with the greatest disease burden. The overall aim of this study was therefore to assess the prevalence and associated factors of stillbirth among women who delivered in public hospitals of Wolaita zone.

## Methods and materials

### Study design and period

A facility-based cross-sectional study design was conducted in health facilities in the Wolaita zone from August 2019 to September 2019.

### Study setting

Wolaita Zone is one of the 14 Zones in southern nations, nationalities, and peoples' regional (SNNPR) government; at a distance of 380 km from Addis Ababa, the capital city of Ethiopia. This zone consists of 16 Woredas/districts and 6 city administrations. The projected total population was 2,085,727 with (1,022,006) males and (1,063,721) females. There are about 72 health centers, 4 primary hospitals, 2 private general hospitals, and 1 teaching and referral hospital. According to the Wolaita zone health department health management information system (HMIS) report, 2018, stillbirth was 160 per 45582 live births, from which mostly about 95% were reported from public hospitals.

### Population

All deliveries of women in the public and private health facilities of Wolaita Zone was considered as source population. And all deliveries of women in the selected public and private health facilities were taken as the study population.

### Sample size determination

**1st objective.** The sample size is calculated for the 1st objective by using the single population proportion formula taken from the study done on incidence and determinants of stillbirth among women who gave birth in Jimma University specialized hospital, Ethiopia [6] with the prevalence of 8% of stillbirth.

$$n = \frac{(Z1 - \alpha/2)^2 * P * (1 - P)}{d^2}$$

Where
n = estimated Sample Size
Z1-α/2=the standard normal value corresponding to the desired level of confidence 95% corresponds to the value of 1.96.
d = margin of sampling error tolerated 5% =0.05

P = is an estimate of the prevalence rate for the population (an assumption that stillbirth deliveries among laboring women in the study area)

$$n = \frac{(1.96)^2 * 0.08 * (1 - 0.08)}{0.05^2} = 113$$

So by considering a 5% none—response rate, the total sample required was 119.

**2nd objective.** The sample size for the second objective is calculated using OpenEpi statistical software version 3.03 for factors associated with stillbirth among delivered women from previous studies (Table 1).

The sample size calculated for the second objective is higher than the sample size calculated for the first objective. Therefore, the largest sample size 737 is used as the final sample size for this study.

## Sampling techniques

The stratified sampling technique was used to select the study participants. A simple random sampling method was used to select the required health facilities in the Wolaita zone. From 72 health centers, 4 primary hospitals, 2 private general hospitals, and 1 teaching and referral hospital, two primary hospitals (BALE and BITENA primary hospitals) and one referral hospital (WSU teaching and referral hospital), and seven health centers namely Sodo, Bodit, Badessa, Dimtu, Gununo, Gasuba, and Humbo health centers which provides routine delivery services for laboring women were selected randomly and the required sample size was allocated to selected health facilities proportionally.

## Measurement

The structured questionnaire adapted from similar studies was used [11, 12]. It is divided into five parts. The first section inquired about personal data, including age, occupation, ethnicity, religion, and educational level. The second part elicited information about Obstetric and Reproductive history. The third section was Health service access variables. The fourth section inquired about Behavioral history. The fifth part elicited information about Maternal-fetal factors.

## Data collection process

Eight diploma graduate midwifery nurses as data collectors and 1 BSc midwifery and 1 health officer supervisor who fluently speak Amharic and Wolaita language were recruited. The questionnaire was prepared in English and then translated into Amharic and Wolaita language and back-translated to English by language experts to check its consistency. Two days of in-depth training was given for data collectors on the overview of research ethics, data collecting tools, and how to fill out the questionnaire. The interviews were conducted after childbirth and before discharge from the facility.

**Table 1. Sample size determination of 2nd objective.**

| S. no | Variables | CI | Power | % of controls exposed | References | Ratio of controls to cases | AOR | Sample Size | Non-response | Total sample size |
|---|---|---|---|---|---|---|---|---|---|---|
| 1 | Absence of complication | 95% | 80 | 30.3 | 17 | 1:1 | 0.1 | 78 | 4 | 82 |
| 2 | Referral from other facility | 95% | 80 | 30.3 | 17 | 1:1 | 0.3 | 166 | 8 | 174 |
| 3 | ANC follow-up | 95% | 80 | 2.9 | 12 | 1:1 | 0.03 | 702 | 35 | 737 |

## Data analysis

Data were edited, coded, and entered into Epidata version 3.1 and exported to SPSS 20 statistical software for analysis. After cleaning data for inconsistencies and missing values in SPSS, descriptive statistics were done. Missing data analysis was conducted by assuming data was missing completely at random (MCAR). Bivariate analysis was done to determine the association between each independent variable and the outcome variable. Before building a bivariable binary logistic regression model, variables significant in other studies and having biological plausibility were selected. In bivariable binary logistic regression, all predictor variables with a p-value of less than 0.25 were identified and entered into a multivariable logistic regression model.

Then a multivariable logistic regression model using a backward stepwise selection method at P value< 0.05 and AOR with 95% CI were used to measure the degree of association between independent variables and the outcome variable. Finally, the result was presented by texts, tables, charts, and figures.

## Data quality control

Two days of in-depth training was given to data collectors. Data collection was supervised by supervisors and the principal investigator. A pretest was conducted on 5% of the sample size in Bedessa Health Center, Wolaita zone, southern Ethiopia. Data were cleaned and checked for completeness daily.

## Ethical considerations

Ethical clearance was secured from the ethical clearance committee of the Wolaita Sodo University, College of Health Science and Medicine. The concerned officials at all levels were informed to get the assurance of the study. The purpose, objectives, and importance of the study were explained and both written and verbal informed consent was secured from each participant. The participant was reassured about the loss of the baby and further advice was given. They were told that documents will be kept confidential and have the right to refuse participation totally at any time if they were not comfortable.

# Results

## The socio-demographic characteristics

Of a total of 737 women, 725 women have successfully answered the survey question, which makes the response rate 98.3%. Of these respondents, 571 (78.8%) were grouped into ages20-34 years. The mean age of the respondents was 26.8 (SD± 4.9). The majority of the study participants were followers of the protestant religion 387 (53.4%) and about 292 (40.3%) of the women were housewives (Table 2).

## Obstetric and reproductive characteristics

Of the total study participants, 351 (48.4%) had 2–3 children which are alive and 429 (59.2%) of women had 2–4 pregnancies. The majority of the study participants, 699 (96.4%) were at37-42 weeks of gestational age. The majority of the study participants had ANC follow-up 622 (85.8%) and about 379 (60.9%) of the study participants had four and above ANC visits (Table 3).

**Table 2. Frequency distributions of socio-demographic characteristics of women who gave birth in public hospitals of Wolaita zone, SNNPR, Ethiopia, 2019 (n = 725).**

| Characteristics (n = 725) | | Frequency | Percent |
|---|---|---|---|
| Age of mother | <20 | 96 | 13.2 |
| | 20–34 | 571 | 78.8 |
| | > = 35 | 58 | 8.0 |
| Residence | Rural | 376 | 51.9 |
| | Urban | 349 | 48.1 |
| Religion | Orthodox | 228 | 31.4 |
| | Catholic | 92 | 12.7 |
| | Protestant | 387 | 53.4 |
| | Muslim | 15 | 2.1 |
| | Others* | 3 | 0.4 |
| Educational status of women | Unable to write and read | 206 | 28.4 |
| | Primary | 226 | 31.2 |
| | Secondary | 176 | 24.3 |
| | Diploma and above | 117 | 16.1 |
| Occupation | Housewife | 292 | 40.3 |
| | Farmer | 44 | 6.1 |
| | Merchant | 106 | 14.6 |
| | Student | 97 | 13.4 |
| | Daily laborer | 30 | 4.1 |
| | Government employee | 156 | 21.5 |
| Ethnicity | Wolaita | 662 | 91.3 |
| | Amhara | 18 | 2.5 |
| | Gurage | 15 | 2.1 |
| | Others** | 30 | 4.1 |
| Monthly income | <500 | 110 | 15.2 |
| | 500–1500 | 235 | 32.4 |
| | > = 1500 | 201 | 27.7 |
| | Don't know | 179 | 24.7 |
| Age at first pregnancy | <20 | 185 | 25.5 |
| | 20–24 | 452 | 62.3 |
| | > = 25 | 88 | 12.2 |

* = 7th day Adventist

** = Gamo, Gofa, Kambata, Silte

### Health service access variables

Of the total study participants, 329 (45.4%) used cars as a mode of transportation to the health facility. The majority of the study participants, 475 (65.5%) of the study participants came to the hospital with a referral. Of the total study participants, 602 (83.0%) reached the health facility in less than 30 minutes (Table 4).

### Lifestyle behavioral variables

Of the total study participants, 692 (95.4%) had no history of alcohol intake in the past 1year and about 719 (99.2%) of the study participants didn't currently smoke a cigarette. Almost all of the study participants didn't chew chat currently 719 (99.2%) (Table 5).

**Table 3. Frequency distributions of obstetric and reproductive characteristics of women who gave birth in public hospitals of Wolaita zone, SNNPR, Ethiopia, 2019 (n = 725).**

| Characteristics (n = 725) | Category | Frequency | Percent |
|---|---|---|---|
| Number of children ever born (n = 725) | 1 | 230 | 31.7 |
| | 2–3 | 351 | 48.4 |
| | 4–5 | 110 | 15.2 |
| | > = 6 | 34 | 4.7 |
| Number of pregnancies(n = 725) | 1 | 215 | 29.7 |
| | 2–4 | 429 | 59.2 |
| | > = 5 | 81 | 11.1 |
| Gestational age in weeks (n = 725) | <37 weeks | 12 | 1.7 |
| | 37–42 weeks | 699 | 96.4 |
| | >42 weeks | 14 | 1.9 |
| ANC visit for current pregnancy (n = 725) | Yes | 622 | 85.8 |
| | No | 103 | 14.2 |
| Number of ANC visits(n = 622) | 1 | 21 | 3.4 |
| | 2–3 | 222 | 35.7 |
| | > = 4 | 379 | 60.9 |
| Time for first ANC visit (n = 622) | <3 | 42 | 6.8 |
| | 3–5 | 420 | 67.5 |
| | > = 6 | 160 | 25.7 |
| Iron/folic during ANC visit for current pregnancy(n = 622) | Yes | 552 | 76.1 |
| | No | 70 | 9.7 |
| Admission during Pregnancy and labor (n = 725) | Yes | 95 | 13.1 |
| | No | 630 | 86.9 |
| Diagnosis (n = 95) | Malaria | 45 | 47.4 |
| | Anemia | 16 | 16.8 |
| | Pregnancy-induced hypertension | 20 | 21.1 |
| | Others | 14 | 14.7 |
| Complications during pregnancy and labor | Yes | 93 | 12.8 |
| | No | 632 | 87.2 |
| Excessive bleeding per vagina (n = 725) | Yes | 12 | 1.7 |
| | No | 713 | 98.3 |
| Labor stay in hours (n = 725) | <12 | 360 | 49.7 |
| | > = 12 | 365 | 50.3 |
| About current pregnancy (n = 725) | Wanted | 595 | 82.1 |
| | Unwanted | 130 | 17.9 |

* = urinary tract infection, typhoid fever

## Maternal and fetal characteristics of deliveries

Of the total study participants, 623 (85.9%) had no history of stillbirth. Most of the study participants 531(73.2%) delivered without complication. About 520 (71.7%) of the study participants gave birth with a spontaneous vaginal delivery as a mode of delivery (Table 6).

## Prevalence of stillbirth

The overall prevalence of stillbirth was 8.7% with (95% CI: 6.5–10.8) among study participants. Of the total study participants, 63 (8.7%) women experienced stillbirth with a stillbirth rate of 87 per 1000 live birth.

**Table 4. Frequency distributions of health service access characteristics of deliveries attended in public hospitals in Wolaita zone, SNNPR, Ethiopia, 2019 (N = 725).**

| Characteristics (N = 725) | Category | Frequency | Percent |
|---|---|---|---|
| Mode of transportation | On foot | 249 | 34.3 |
| | Motorcycle | 54 | 7.4 |
| | Car | 329 | 45.4 |
| | Bajaj(taxi) | 93 | 12.8 |
| Come to hospital | With referral | 475 | 65.5 |
| | Without referral | 250 | 34.5 |
| Time to reach the hospital | <30 minutes | 602 | 83.0 |
| | > = 30 minutes | 123 | 17.0 |

### Factors associated with stillbirth

The multivariable logistic regression analysis was done to control confounders of associations with the outcome variable. After adjusting for several significant covariates associated in the bivariate analysis, residence, complication during pregnancy and labor, history of stillbirth, number of pregnancy, and type of delivery was independently associated factors of stillbirth (Table 7).

The odds of stillbirth were 2.57 (95% CI: 1.23–5.40) times higher among rural residents when compared to urban residents. The study participants with a complication during pregnancy and labor were associated with stillbirth with (AOR6.23; (95% CI: 2.67–14.58)) compared to those with no complication during pregnancy and labor. This study also revealed that the study participants who had two to four pregnancies were 3.82 times more likely to be associated with stillbirth (AOR (3.82; 95% CI: 1.17–12.47)) compared to those with one pregnancy. Those study participants with a history of stillbirth were 6.89 times associated with stillbirth with AOR (5.91; 95% CI: 2.57–13.57) compared to their counterparts. The odds of stillbirth was 7.13 (95% CI: 2.71–18.73) times higher among complicated deliveries when compared to normal deliveries (Table 7).

### Discussion

This study assessed the prevalence of stillbirth and associated factors among deliveries attended in public hospitals in the Wolaita zone in Southern Ethiopia. Accordingly, the prevalence of stillbirth among women who gave birth in public hospitals in this study was found to be 8.7% (95% CI:6.5–10.8). It also showed that residence, number of pregnancy, complications

**Table 5. Frequency distributions of lifestyle behavioral characteristics of women who gave birth in public hospitals of Wolaita zone, SNNPR, Ethiopia, 2019 (n = 725).**

| Characteristics (n = 725) | Category | Frequency | Percent |
|---|---|---|---|
| Alcohol intake in the past 1 year 1 year | Yes | 33 | 4.6 |
| | No | 692 | 95.4 |
| Currently smoke cigarette | Yes | 6 | 0.8 |
| | No | 719 | 99.2 |
| Chewing chat currently | Yes | 6 | 0.8 |
| | No | 719 | 99.2 |
| Used any herbal medication | Yes | 39 | 5.4 |
| | No | 686 | 94.6 |

**Table 6. Frequency distributions of maternal and fetal characteristics of women who gave birth in public hospitals of Wolaita zone, SNNPR, Ethiopia, 2019 (n = 725).**

| Characteristics (n = 725) | Category | Frequency | Percent |
|---|---|---|---|
| History of stillbirth | Yes | 102 | 14.1 |
| | No | 623 | 85.9 |
| Type of delivery | Normal | 531 | 73.2 |
| | Complicated | 194 | 26.8 |
| Mode of delivery | SVD | 520 | 71.7 |
| | CS | 111 | 15.3 |
| | Instrumental | 94 | 13.0 |
| Congenital anomalies | Yes | 18 | 2.5 |
| | No | 707 | 97.5 |
| Fetal presentation | Cephalic | 638 | 88.0 |
| | Breach | 54 | 7.4 |
| | Shoulder | 18 | 2.5 |
| | Others | 15 | 2.1 |
| Cord prolapsed | Yes | 48 | 6.6 |
| | No | 677 | 93.4 |

* = face presentation, brow presentation, limb presentation

**Table 7. Bivariate and multivariate logistic regression analysis of factors associated with stillbirth in public hospitals of Wolaita zone, SNNPR, Ethiopia, 2019.**

| Variables | Category | Stillbirth | | COR(95% CI) | AOR(95% CI) |
|---|---|---|---|---|---|
| | | Yes | No | | |
| Residence | Rural | 43(10.8) | 356(89.2) | 1.85(1.22–3.69)** | 2.57(1.23–5.40)* |
| | Urban | 20(6.1) | 306(93.9) | 1 | 1 |
| Maternal age | <20 | 6(6.2) | 90(93.8) | 1 | 1 |
| | 20–34 | 46(8.1) | 525(91.9) | 1.31(0.83–1.95) | 0.23(0.04–1.34) |
| | > = 35 | 11(19) | 47(81) | 3.51(0.92–6.82) | 0.15(0.04–0.55) |
| Complication | Yes | 36(38.7) | 57(61.3) | 14.15(8.02–24.98)*** | 6.23(2.67–14.58)*** |
| | No | 27(4.3) | 605(95.7) | 1 | 1 |
| Referral | With referral | 14(5.6) | 236(94.4) | 0.52(0.28–0.95) | 0.65(0.28–1.52) |
| | Without referral | 49(10.3) | 426(89.7) | 1 | 1 |
| No of pregnancy | 1 | 11(6.3) | 164(93.7) | 1 | 1 |
| | 2–4 | 36(9.6) | 340(90.4) | 1.58(1.41–5.12)** | 3.82(1.17–12.47)* |
| | > = 5 | 16(9.2) | 158(90.8) | 1.51(0.52–5.14) | 2.07(0.43–9.86) |
| History of stillbirth | Yes | 29(28.4) | 73(71.6) | 6.88(3.96–11.95)*** | 5.91(2.57–13.57)*** |
| | No | 34(5.5) | 589(94.5) | 1 | 1 |
| Mode of delivery | SVD | 18(3.5) | 502(96.5) | 0.17(0.09–0.36) | 1.43(0.51–4.01) |
| | CS | 29(26.1) | 82(73.9) | 1.72(0.87–3.42) | 2.59(0.76–8.81) |
| | Instrumental | 16(17.0) | 78(83) | 1 | 1 |
| Type of delivery | Normal | 14(2.6) | 517(97.4) | 1 | 1 |
| | Complicated | 49(25.3) | 145(74.7) | 12.48(6.70–23.24)*** | 7.13(2.71–18.73)*** |

* = P<0.05,

** = P<0.01,

*** = P<0.001

during pregnancy and labor, history of stillbirth, and type of delivery were independently associated factors of stillbirth among deliveries in this study.

The prevalence of stillbirth in this study showed a comparable proportion with the study conducted in the Amhara region, Ethiopia (8.5%) [6]. Besides, it is in line with the study done in Negest Elene Mohammed general hospital in Hosanna town, South Ethiopia (8.6%) [13]. However, the prevalence in this study was higher than the report of the Ethiopian demographic and health survey, SNNPR (2.0%) [8]. This variation might be due to the difference in the setting in which the study was conducted; the current study was conducted only in a hospital where most of the cases were referred from the peripheral health facility after delays that complicate labor.

In this study, the place of residence is significantly associated with stillbirth. Women from rural residences experienced nearly three times more still births compared to urban women. This is in line with the study done in Ethiopia, Hosanna town, SNNPR, and North Wollo zone, Northeast Ethiopia [2, 13, 14]. This is explained by the fact that women residing in a rural areas may not take action early due to poor health access and limited information about pregnancy, labor, and delivery. Besides, there might be an occurrence of delays among women residing in rural areas.

The number of pregnancies was also significantly associated with stillbirth in the current study. Multigravida women were around four times more likely to have stillbirth compared to women with primigravida. A similar finding was reported in the study conducted in the North Wollo zone and suhul hospital Shire Tigray [14, 15]. It might be also the negligence of women to take health services promptly after the first pregnancy. Besides, it might also be due to poor socio-economic status that leads to the sharing of food with family members.

Complication during pregnancy and labor was also significantly associated with stillbirth. Women with a complication during pregnancy and labor were six times more likely to encounter stillbirth compared to those women without complications. This is in line with the study done in Hosanna town and the North Wollo zone [13, 14]. It could be explained that, if untreated and not mentioned by the mother early, it could end the life of the mother not only the fetus. Failure to recognize signs of complications is one of the barriers to the delay in deciding to seek care [16].

The study also revealed that a history of stillbirth was the factor significantly associated with stillbirth. Women with a history of stillbirth were nearly six times more likely to encounter stillbirth than their counterparts. A similar finding was reported in the study conducted in Northwest Ethiopia showing a history of stillbirth as a predictor of stillbirth [17]. This might be because women with a history of stillbirth regarded as being at high risk of another stillbirth and due to some of the poor obstetric health conditions are recurrent [18].

The type of delivery was also significantly associated with stillbirth. In this study, women with complicated deliveries were seven times more likely to encounter stillbirth compared to those women with normal deliveries. This finding goes in line with the finding of the study conducted in Jimma university specialized hospital [11]. This is explained by the fact that most of the women from lower-level health facilities were referred to hospitals when they get complicated. Also, deliveries other than spontaneous vaginal delivery have a high risk of trauma on a baby that leads to the death of a baby before birth.

## Strength of the study

The strength of this study was that it included primary hospitals and referral hospital. The data were collected by health professionals who were involved in similar studies before.

### Limitation of the study

Some limitations of this study were; firstly, the information used in this study was based on self-report. There may be some bias in the reporting, particularly around some of the sensitive issues that were incorporated, for example, history of stillbirth, congenital anomaly, etc. So that, to address these issues confidentiality was strictly maintained for study participants and the data collectors were oriented on the collection of data. Secondly, age data were exposed to recall bias since the women didn't know the exact date of last birth and date of marriage.

## Conclusion

The prevalence of stillbirth among women who gave birth in public hospitals in the Wolaita zone was found to be high compared to national and regional figures in the study area. Residence, number of pregnancy, complications during pregnancy and labor, history of stillbirth, and type of delivery were independent factors affecting stillbirth. The result of this study remarks that improvements in the quality of maternal health services require strict attention.

## Recommendation

Eventually, based on our findings, the following recommendations are forwarded:

➢ The federal government and regional government should strengthen communication and discussion with a grass-root level health facility to promote women to use maternal and health care services.

➢ The zonal health department and other concerned bodies should focus on the implementation of the strategies and policies causing stillbirth.

➢ The health professionals in the hospital diagnose and treat the complication during pregnancy and labor early.

➢ Finally, we recommend researchers to emphasize on conducting qualitative researches that identify factors which that may affect stillbirth.

## Supporting information

**S1 Data.**
(SAV)

**S1 File.**
(DOCX)

## Acknowledgments

We would like to thank the Ministry of Education, Wolaita Sodo University, and Addis Ababa University for their technical support. Next, we would like to acknowledge, Wolaita Sodo teaching and referral hospital, Christian generalized hospital, data collectors, supervisors, study participants, and all who gave their hands in the study directly or indirectly without whom the research would not be done.

## Author Contributions

**Conceptualization:** Jegnaw Wolde, Dereje Haile, Kebreab Paulos, Asrat Chernet Adeko.

**Data curation:** Jegnaw Wolde, Dereje Haile, Mihiretu Alemayehu.

**Formal analysis:** Jegnaw Wolde, Dereje Haile, Kebreab Paulos, Mihiretu Alemayehu.

**Funding acquisition:** Jegnaw Wolde, Dereje Haile.

**Investigation:** Jegnaw Wolde, Dereje Haile, Kebreab Paulos, Mihiretu Alemayehu, Asaminew Ayza.

**Methodology:** Jegnaw Wolde, Dereje Haile, Kebreab Paulos, Mihiretu Alemayehu, Asrat Chernet Adeko, Asaminew Ayza.

**Project administration:** Jegnaw Wolde, Dereje Haile, Kebreab Paulos, Asrat Chernet Adeko, Asaminew Ayza.

**Resources:** Jegnaw Wolde, Dereje Haile, Asrat Chernet Adeko, Asaminew Ayza.

**Software:** Jegnaw Wolde, Dereje Haile, Mihiretu Alemayehu, Asrat Chernet Adeko.

**Supervision:** Jegnaw Wolde, Dereje Haile, Kebreab Paulos, Mihiretu Alemayehu, Asrat Chernet Adeko, Asaminew Ayza.

**Validation:** Jegnaw Wolde, Dereje Haile, Kebreab Paulos, Mihiretu Alemayehu, Asrat Chernet Adeko, Asaminew Ayza.

**Visualization:** Jegnaw Wolde, Dereje Haile, Mihiretu Alemayehu, Asrat Chernet Adeko, Asaminew Ayza.

**Writing – original draft:** Jegnaw Wolde, Dereje Haile, Kebreab Paulos, Asaminew Ayza.

**Writing – review & editing:** Jegnaw Wolde, Dereje Haile, Kebreab Paulos, Asaminew Ayza.

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
