## [Decision Letter · Decision Letter 0]

12 Jan 2022

PONE-D-21-20973Prevalence of Stillbirth and associated factors among womens’ deliveries attended in Health Facilities in Southern Ethiopia.PLOS ONE

Dear Dr. Haile,

Thank you for submitting your manuscript to PLOS ONE. After careful consideration, we feel that it has merit but does not fully meet PLOS ONE’s publication criteria as it currently stands. Therefore, we invite you to submit a revised version of the manuscript that addresses the points raised during the review process.

The manuscript has been evaluated by three reviewers, and their comments are available below.

The reviewers have raised a number of concerns that need attention. They request additional information on methodological aspects of the study, the statistical analyses and the manner in which the discussion and conclusions are supported by the results.

Could you please revise the manuscript to carefully address the concerns raised?

We look forward to receiving your revised manuscript.

Kind regards,

Sebastian Shepherd

Associate Editor

PLOS ONE

Journal Requirements:

2. Please include additional information regarding the survey or questionnaire used in the study and ensure that you have provided sufficient details that others could replicate the analyses. For instance, if you developed a questionnaire as part of this study and it is not under a copyright more restrictive than CC-BY, please include a copy, in both the original language and English, as Supporting Information. If the original language is written in non-Latin characters, for example Amharic, Chinese, or Korean, please use a file format that ensures these characters are visible.

3. Please state whether you validated the questionnaire prior to testing on study participants. Please provide details regarding the validation group within the methods section.

4. Please amend your current ethics statement to address the following concerns: Please explain why written consent was not obtained, how you recorded/documented participant consent, and if the ethics committees/IRBs approved this consent procedure.

5. Thank you for stating the following in the Acknowledgments / Funding Section of your manuscript: 

The source of funding for this research was covered by the Ministry of Education in collaboration with Wolaita Sodo University for resources and data collection.  

The author(s) received no specific funding for this work

7. We noticed you have some minor occurrence of overlapping text with the following previous publication(s), which needs to be addressed:

- https://researchonline.lshtm.ac.uk/id/eprint/4655794/1/2020_EPH_PhD_Blencowe_H-Copy.pdf

In your revision ensure you cite all your sources (including your own works), and quote or rephrase any duplicated text outside the methods section. Further consideration is dependent on these concerns being addressed.

Reviewers' comments:

Reviewer's Responses to Questions

**Comments to the Author**

1. Is the manuscript technically sound, and do the data support the conclusions?

Reviewer #1: No

Reviewer #2: Yes

Reviewer #3: Partly

2. Has the statistical analysis been performed appropriately and rigorously? 

Reviewer #1: No

Reviewer #2: Yes

Reviewer #3: No

3. Have the authors made all data underlying the findings in their manuscript fully available?

Reviewer #1: Yes

Reviewer #2: Yes

Reviewer #3: Yes

4. Is the manuscript presented in an intelligible fashion and written in standard English?

Reviewer #1: No

Reviewer #2: Yes

Reviewer #3: No

5. Review Comments to the Author

Reviewer #1: First of all, I thank you for inviting to review manuscript "Prevalence of Stillbirth and associated factors among womens’ deliveries attended in Health Facilities in Southern Ethiopia (PONE-D-21-20973)" for the journal of PLOS ONE. And I have the following comments for the authors that the manuscript might benefit from.

General comments:

- There are so many grammatical and punctuation errors in the whole document and it needs language edition.

- Abbreviations should state with its full at the first time and this is not in your work and also have no found in the list of abbreviation section (E.g.: SNNPR).

- Tables are not in a standard format.

- Some of your statements have no source (for instance, According to the WHO definition, Stillbirth is a baby born with no signs of life at or after 28 weeks’ gestation) and many more.

- The organization of introduction is not appropriate (it is better organized either worldwide�continent� country� state/region OR considering the chronological order of references).

Abstract:

- The abstract section is not well stated, for instance, it is better that the background and objective should state with two or three lines as one paragraph.

- Authors stated that “Stratified sampling techinique was used to select 737 mothers.” What was the criterion that you stratified and which mother you select?

- Bivariate and multiple logistic regression analysis…. Be rewrite as bi-variable and multi-variable logistic….and it is true to for all in the body of the manscript.

- The statement “…..odds ratios at a 95% confidence interval with P-value <0.05 were considered to declare a result as statistically significant…” is redundancy that is 95% CI and p-value 0.05 are the same and writing of “p-value<0.05” is informal.

- “Being rural residence ……. with stillbirth” is a very long statement and did not well expressed. For instance, does rural residence a variable of level of variable? Some to others

- Conclusion and recommendation are not well stated and it needs amendment.

Introduction:

-Paragraph 4 line 2-3: authors stated that “There is a problem with appropriate data on stillbirth in developing countries such as Ethiopia.” What do mean by this? I belief there is a worldwide appropriate data (EDHS-2016 and Mini EDHS-2019) and this is is not correct statement.

-Paragraph 4 line 5-6: “The data generally available in low and middle-income countries to inform the cause of death often includes only a limited obstetric and/or reproductive history” sorry to say, I am not clear with this statement. Which data that inform the causes of death and for which death?

-Lastly what is the new contribution of this manuscript to scholars and scientists worldwide? It is not well stated and showed the gap that fill full by this work.

Methods, results, discussion, conclusion and recommendations:

- What do mean by SNNPR, HMIS …? It is better to state with its long form at the first time?

- Why you consider Wolaita Zone among all of 14 Zones? What do mean by Zones (what is the definition of Zone in your country)?

- “According to the Wolaita zone health department…” this statement is miss placed, it is better to move somewhere as this is not indicate the study setting.

- Your sampling method was stratified, but you calculated the sample size based on simple random sampling method. Thus, the way you calculate the sample size is wrong and better to recalculate using stratified methods.

- What is/are the nature of variable(s) that you stratified your study population?

- A lot of grammatical errors are found in the manuscript. For instance, see this statement “Eight diploma graduate midwifery nurses as data collectors and 1 BSc midwifery and 1 health officer supervisor who speak Amharic and Wolaitegna very well were recruited.” What do mean by? Sorry, I can’t understand it!

- As far as I know “Bivariate analysis” and “bi-variable analysis” very different and I think you did not perform a bivariate analysis rather you did bi-variable. Thus, correct “bivariate and multivariate” as “bi-variable and muli-variable”

- You did not indicate the model that you used in this manuscript and it is better to briefly state the model in the method section.

- What is your response/outcome variable and how you define it? The same for your independent variables. It is better to have sub sections of variables in the study and it should explain the definition, units/levels of both outcome and explanatory variables.

- Some where you stated that “571 (78.8%) were grouped into age 20-34 years” and the productive age range of the country Ethiopia is from 15 to 49 years. But your sample is not representative as about 80% is from 20-34 years and these misslead your result and conclusion. And again, the mean age range of the population is from 21.9 to 31.7 years, which indicates that the sample is not well representative of the population. By the way what do mean by SD±4.9 from “The mean age of the respondents was 26.8 (SD+ 4.9).”?

- You stated that “The overall prevalence of stillbirth was 8.7%” does it prevalence or---? Because, (63/725)*100%=8.689% is not prevalence. Thus, first define what do mean by prevalence and you should recalculate the prevalence of stillbirth correctly otherwise your conclusion is misleading Zonal administrations!!!!!!

- The result section did not well stated and elaborated.

- You stated that “The prevalence of stillbirth in this study showed a comparable proportion with the study conducted in the Amhara region, Ethiopia (8.5%)(10).” But Lakew et.al, 2017 stated that the proportion of stillbirth is 8.5% (218 out of 2555 samples) whereas the prevalence is 85 per 1000 total births. Thus, your comparison and definition of prevalence is totally wrong.

- Consider the levels of religion that Orthodox (228) and Others (3), do you think that this frequencies are comparable and have no effect on parameter estimation? And that is why many of your predictors are not statistically significant in your model.

- Monthly income is lebaled as <500, 500-1500 and <= 15000, what is your evidence to label like this? How you measure the monthly income of mothers as the majority were have no constant monthly income (housewife plus farmers =334).

- You conclusion and recommendations are not we organized and it needs amendments.

- Finally, Tables are not well organized, some texts are not visible and all tables needs rearrangements.

Reviewer #2: Dear all,

This is an important manuscript. My main point relates to how the results are being discussed. The results states that stillbirth correlates to being resident in rural areas, frequent pregnancies, and complications during pregnancy and labour. In the discussion section you tell the reader why this is the case, but without telling how this could be solved, implemented as strategies, policies, information, respectful care etc. Later in the discussion section I read recommendations and find what I search for in that section. I strongly recommend that you edit the discussion section and make the recommendations come consequently when the readers read the discussion and get the explanation of why stillbirth exists for instance in rural areas. After that the reader wish to hear your recommendation and solution of the problem, and not in a separate section further down. Now you leave the reader with a ...and what then...?? kind of feeling. It will be much better if you structure the discussion like that according to me.

You have some errors in space between words. Check that out please.

Good luck with this manuscript.

Reviewer #3: Stillbirths is indeed an important problem especially in low and middle income countries (LMICs). It is therefore important to assess the prevalence and associated factors of stillbirths in these settings.

The authors aim to assess the prevalence and associated factors among deliveries attended in public hospitals of Wolaita zone, southern Ethiopia.

However, the manuscript in its current form needs some major revisions before it can be suitable for publication.

1. The methods need more detail added such as how the authors determined the number of institutions to be included in the sample. The derivation of total number of participants required was clearly shown but not the institutions.

2. The statistical method used to calculate the confidence interval for the proportion of stillbirths 8.7% also need to be clearly explained (was any bootstrap method used?)

3. It is not clear how the initial set of predictors/independent variables were selected for inclusion in the bivariate analyses and subsequent backward elimination models. The authors may consider methods such as Directed Acyclic Graphs (DAGs) for determining the set of confounders to include in the multivariable model.

4. Under the Data Analysis section, the authors mention that descriptive statistics were done after cleaning for inconsistencies and missing values. It would be good to mention what methods for handling missing values were employed.

5. The data analysis section also mentions that "the data was presented by texts, tables, charts,and figures". I did not come across any figures. Maybe the authors forgot to attach them with this submission.

6. The authors may consider moving the section on Data quality control to come before the Data Analysis section.

6. PLOS authors have the option to publish the peer review history of their article (what does this mean?). If published, this will include your full peer review and any attached files.

Reviewer #1: No

Reviewer #2: No

Reviewer #3: No

---

## [Author Response · Author response to Decision Letter 0]

22 Feb 2022

Review Comments to the Author

Reviewer #1: First of all, I thank you for inviting to review manuscript "Prevalence of Stillbirth and associated factors among womens’ deliveries attended in Health Facilities in Southern Ethiopia (PONE-D-21-20973)" for the journal of PLOS ONE. And I have the following comments for the authors that the manuscript might benefit from.

General comments:

- There are so many grammatical and punctuation errors in the whole document and it needs language edition.

Response: thank you for your comment; we have corrected it in the revised manuscript. 

- Abbreviations should state with it’s full at the first time and this is not in your work and also have no found in the list of abbreviation section (E.g.: SNNPR).

Response: thank you for your suggestion, we have corrected and highlighted it in the whole documents of the revised manuscript.

- Tables are not in a standard format.

Response: thank you for your comment, we have corrected it.

- Some of your statements have no source (for instance, According to the WHO definition, Stillbirth is a baby born with no signs of life at or after 28 weeks’ gestation) and many more.

Response: we have added the citation.

 State/region OR considering the chronological order of references). country�continent�- The organization of introduction is not appropriate (it is better organized either worldwide

Response: we have rearranged the introduction section according to your suggestion.

Abstract:

- The abstract section is not well stated, for instance, it is better that the background and objective should state with two or three lines as one paragraph.

Response: we have corrected and highlighted in the revised manuscript.

- Authors stated that “Stratified sampling techinique was used to select 737 mothers.” What was the criterion that you stratified and which mother you select?

Response: thank you for your meticulous observation; and sorry for what we have made an error, we have used systematic random sampling to select the study participants.

- Bivariate and multiple logistic regression analysis…. Be rewrite as bi-variable and multi-variable logistic….and it is true to for all in the body of the manscript.

Response: thank you for your suggestion; we have corrected it according to your suggestion.

- The statement “…..odds ratios at a 95% confidence interval with P-value <0.05 were considered to declare a result as statistically significant…” is redundancy that is 95% CI and p-value 0.05 are the same and writing of “p-value<0.05” is informal.

Response: thank you for your comment, we have corrected it

- “Being rural residence ……. with stillbirth” is a very long statement and did not well expressed. For instance, does rural residence a variable of level of variable? Some to others

Response: thank you for your comment, we have corrected “being residence” as residence. 

- Conclusion and recommendation are not well stated and it needs amendment.

Response: thank you for your comment; we have amended it as much as possible.

Introduction:

-Paragraph 4 line 2-3: authors stated that “There is a problem with appropriate data on stillbirth in developing countries such as Ethiopia.” What do mean by this? I belief there is a worldwide appropriate data (EDHS-2016 and Mini EDHS-2019) and this is is not correct statement.

Response: thank you for your comment, we have corrected it. Even though we have a dearth of evidences on the prevalence and its predictors of stillbirth among delivered mothers, we have limited evidences in this study area.

-Paragraph 4 line 5-6: “The data generally available in low and middle-income countries to inform the cause of death often includes only a limited obstetric and/or reproductive history” sorry to say, I am not clear with this statement. Which data that inform the causes of death and for which death?

Response: thank you for your comment, we have amended and highlighted it in the revised manuscript.

-Lastly what is the new contribution of this manuscript to scholars and scientists worldwide? It is not well stated and showed the gap that fill full by this work.

Response: we have added the novelty of this study in the introduction section of the revised manuscript.

Methods, results, discussion, conclusion and recommendations:

- What do mean by SNNPR, HMIS …? It is better to state with its long form at the first time?

Response: thank you for your comment; we have spelled out when the abbreviation comes first throughout the documents.

- Why you consider Wolaita Zone among all of 14 Zones? What do mean by Zones (what is the definition of Zone in your country)?

Response: zone is a subset of region. Thus, wolaita zone is one of the densely populated zone among 14 zones in the Southern People Nation and Nationality (SNPPR) in Ethiopia.

- “According to the Wolaita zone health department…” this statement is miss placed, it is better to move somewhere as this is not indicate the study setting.

Response: thank you for your suggestion, we have corrected it.

- Your sampling method was stratified, but you calculated the sample size based on simple random sampling method. Thus, the way you calculate the sample size is wrong and better to recalculate using stratified methods.

Response: sorry for an error what we have made, we have followed systematic random sampling to select the study participants rather than stratified random sampling.

- What is/are the nature of variable(s) that you stratified your study population?

Response: sorry again, we have used systematic random sampling to select the study participants.

- A lot of grammatical errors are found in the manuscript. For instance, see this statement “Eight diploma graduate midwifery nurses as data collectors and 1 BSc midwifery and 1 health officer supervisor who speak Amharic and Wolaitegna very well were recruited.” What do mean by? Sorry, I can’t understand it!

Response: thank you for your meticulous observation, we have corrected and highlighted it in the revised manuscript.

- As far as I know “Bivariate analysis” and “bi-variable analysis” very different and I think you did not perform a bivariate analysis rather you did bi-variable. Thus, correct “bivariate and multivariate” as “bi-variable and muli-variable”

Response: thank you for your comment, we have corrected “bivariate” as bi-variable.

- You did not indicate the model that you used in this manuscript and it is better to briefly state the model in the method section.

Response: thank you for your comment, we have used binary logistic regression model for analyses.

- What is your response/outcome variable and how you define it? 

Response; in this study, the outcome variable was stillbirth. We have defined stillbirth as the birth of a baby with no signs of life after 28 weeks of pregnancy recorded by health professional.

The same for your independent variables. It is better to have sub sections of variables in the study and it should explain the definition, units/levels of both outcome and explanatory variables.

Response: thank you for your suggestion; we have included it in the revised manuscript

- Some where you stated that “571 (78.8%) were grouped into age 20-34 years” and the productive age range of the country Ethiopia is from 15 to 49 years. But your sample is not representative as about 80% is from 20-34 years and these miss lead your result and conclusion. And again, the mean age range of the population is from 21.9 to 31.7 years, which indicates that the sample is not well representative of the population. By the way what do mean by SD±4.9 from “The mean age of the respondents was 26.8 (SD+ 4.9).”?

Response: thank you for your comment, by default, majority of the study participants were aged 20-34, but because of majority of the study participants from rural area, they may not know their real age.

- You stated that “The overall prevalence of stillbirth was 8.7%” does it prevalence or---? Because, (63/725)*100%=8.689% is not prevalence. Thus, first define what do mean by prevalence and you should recalculate the prevalence of stillbirth correctly otherwise your conclusion is misleading Zonal administrations!!!!!!

Response: we have corrected it in the revised manuscript

- The result section did not well state and elaborated.

Response: thank you for your comment, we have elaborated it more

- You stated that “The prevalence of stillbirth in this study showed a comparable proportion with the study conducted in the Amhara region, Ethiopia (8.5%)(10).” But Lakew et.al, 2017 stated that the proportion of stillbirth is 8.5% (218 out of 2555 samples) whereas the prevalence is 85 per 1000 total births. Thus, your comparison and definition of prevalence is totally wrong.

Response: thank you for your comment, we have amended and highlighted it in the revised manuscript.

- Consider the levels of religion that Orthodox (228) and Others (3), do you think that this frequencies are comparable and have no effect on parameter estimation? And that is why many of your predictors are not statistically significant in your model.

Response: thank you for your comment; we have not included religion in the final model

- Monthly income is lebaled as <500, 500-1500 and <= 15000, what is your evidence to label like this? 

How you measure the monthly income of mothers as the majority were have no constant monthly income (housewife plus farmers =334).

Response: we have used a study done previously as reference

:

- You conclusion and recommendations are not we organized and it needs amendments.

Response: we have amended it in the revised manuscript

- Finally, Tables are not well organized, some texts are not visible and all tables needs rearrangements.

Response: we have corrected the tables as much as possible.

Reviewer #2: Dear all,

This is an important manuscript. My main point relates to how the results are being discussed. The results states that stillbirth correlates to being resident in rural areas, frequent pregnancies, and complications during pregnancy and labour. In the discussion section you tell the reader why this is the case, but without telling how this could be solved, implemented as strategies, policies, information, respectful care etc. Later in the discussion section I read recommendations and find what I search for in that section. I strongly recommend that you edit the discussion section and make the recommendations come consequently when the readers read the discussion and get the explanation of why stillbirth exists for instance in rural areas. After that the reader wish to hear your recommendation and solution of the problem, and not in a separate section further down. Now you leave the reader with a ...and what then...?? kind of feeling. It will be much better if you structure the discussion like that according to me.

You have some errors in space between words. Check that out please.

Good luck with this manuscript.

Response: thank you for your comments, we have corrected and highlighted it in the revised manuscript 

Reviewer #3: Stillbirths is indeed an important problem especially in low and middle income countries (LMICs). It is therefore important to assess the prevalence and associated factors of stillbirths in these settings.

The authors aim to assess the prevalence and associated factors among deliveries attended in public hospitals of Wolaita zone, southern Ethiopia.

However, the manuscript in its current form needs some major revisions before it can be suitable for publication.

Response: thank you for your meticulous observation.

1. The methods need more detail added such as how the authors determined the number of institutions to be included in the sample. The derivation of total number of participants required was clearly shown but not the institutions.

Response: thank you for your comment; we have elaborated how the institutions were included in the revised manuscript. 

2. The statistical method used to calculate the confidence interval for the proportion of stillbirths 8.7% also need to be clearly explained (was any bootstrap method used?)

Response: thank you for your question, we have used bootstrap method to calculate the confidence interval for the prevalence of stillbirth.

3. It is not clear how the initial set of predictors/independent variables were selected for inclusion in the bivariate analyses and subsequent backward elimination models. The authors may consider methods such as Directed Acyclic Graphs (DAGs) for determining the set of confounders to include in the multivariable model.

Response: thank you for your suggestion, but we have used p-value <=0.25 as cut-off point to select the predictors for multivariable model

4. Under the Data Analysis section, the authors mention that descriptive statistics were done after cleaning for inconsistencies and missing values. It would be good to mention what methods for handling missing values were employed.

Response:

5. The data analysis section also mentions that "the data was presented by texts, tables, charts,and figures". I did not come across any figures. Maybe the authors forgot to attach them with this submission.

Response: thank you for your observation, we have corrected and highlighted it in the revised manuscript.

6. The authors may consider moving the section on Data quality control to come before the Data Analysis section.

Response: thank you for your comment, we have corrected it in the revised manuscript.

---

## [Decision Letter · Decision Letter 1]

23 May 2022

PONE-D-21-20973R1Prevalence of Stillbirth and associated factors among womens’ deliveries attended in Health Facilities in Southern Ethiopia.PLOS ONE

Dear Dr. Haile,

Thank you for submitting your revised manuscript to PLOS ONE. After careful consideration, we feel that it has merit but does not fully meet PLOS ONE’s publication criteria as it currently stands. Therefore, we invite you to submit a revised version of the manuscript that addresses the points raised during the review process.

Please be sure to pay particular attention to the following reviewer concerns: Provide further details on how variables entered into bi-variable regressions were initially selected. Provide further details on missing data and any missing data mechanisms used. Please carefully edit to ensure clarity and that neutral and scientific language is used throughout. Please be consistent about word choice throughout the manuscript, e.g., "women" or "women" rather than "mothers". Adjust the title to better reflect the focus of the manuscript and ensure clarity. Address all other major and minor reviewer comments/concerns. 

We look forward to receiving your revised manuscript.

Kind regards,

Jennifer Manuzak

Academic Editor

PLOS ONE

Reviewers' comments:

Reviewer's Responses to Questions

**Comments to the Author**

1. If the authors have adequately addressed your comments raised in a previous round of review and you feel that this manuscript is now acceptable for publication, you may indicate that here to bypass the “Comments to the Author” section, enter your conflict of interest statement in the “Confidential to Editor” section, and submit your "Accept" recommendation.

Reviewer #2: (No Response)

Reviewer #3: (No Response)

2. Is the manuscript technically sound, and do the data support the conclusions?

Reviewer #2: Yes

Reviewer #3: Partly

3. Has the statistical analysis been performed appropriately and rigorously? 

Reviewer #2: Yes

Reviewer #3: Yes

4. Have the authors made all data underlying the findings in their manuscript fully available?

Reviewer #2: No

Reviewer #3: Yes

5. Is the manuscript presented in an intelligible fashion and written in standard English?

Reviewer #2: No

Reviewer #3: No

6. Review Comments to the Author

Reviewer #2: You have a message, you have improved but jet you need to take another round and improve the manuscript please.

TITLE. I SUGGEST: Prevalence of stillbirth and its associated factors among women giving birth in healthcare facilities in southern Ethiopia

No dot in the title please

OBJECTIVE. I SUGGEST: To assess the prevalence and associated factors of stillbirth among women giving birth at public hospitals of Wolaita zone, southern Ethiopia

RESULTS: I suggest you work with the language to make it less technical. Please replace labor with birth please. A woman give birth …“This finding showed that 8.7% of [95% CI: 6.5-10.8] the stillbirth was among women giving birth in public hospitals. Being rural residence [Adjusted OR = 2.57, 95% CI: 1.23, 5.40], number of pregnancies [Adjusted OR = 3.82, 95% CI: 1.17-12.47], complication during pregnancy and birth [Adjusted OR = 6.23, 95% CI: 2.67-14.58], history of stillbirth [Adjusted OR = 6.89, 95% CI: 2.57-13.57], type of birth [Adjusted OR = 7.13, 95% CI: 2.71-18.73] were found to be factors associated with stillbirth.”

Please choose to call them woman/women and not mothers and please do not mix from paragraph to paragraph: “The prevalence of stillbirth among mothers who gave birth i….

FORMAT: Make sure you do have space between the words: “737 mothers.A pre-tested” The format is still not fixed. It need to look like it should in the journal.

You need to re-work it again. Please take help from someone who can support you to make it look like a manuscript for submission to PLOS ONE. You have a good message to tell but you are jet not there. Please improve more.

Reviewer #3: The authors have attended to a number of the comments raised in the first submission. However, a number of issues remain unaddressed. Generally, the manuscript still require editing as a number of statements are not very clear. For example, under Data collection process, the statement "A pretest was conducted on 5% (37 participants) of the sample size is not selected hospitals in the Wolaita zone and every possible correction was taken" This is not clear and could be completely removed as a similar statement is repeated in the Data quality control section.

On the point 3 I raised the authors have responded that all predictor variables with a p-value less than 0.25 during bivariable regression were entered into a multivariable regression. This is a fair answer but the question still remains how the variables that were entered in the bivariable regressions were selected initially. Some sort of statement justifying these or conceptual framework could be better. Identification of potential confounders can not be left to statistical models entirely.

The authors' response to point 4 is missing. A statement on missing data and any assumed missing data mechanism could help. I assume the the authors carried out complete case analyses which may be valid under certain missing data mechanism assumptions such as MCAR

7. PLOS authors have the option to publish the peer review history of their article (what does this mean?). If published, this will include your full peer review and any attached files.

Reviewer #2: No

Reviewer #3: No

---

## [Author Response · Author response to Decision Letter 1]

1 Jul 2022

Response to reviewers comment

Reviewer #2: You have a message, you have improved but jet you need to take another round and improve the manuscript please.

Response: thank you for your positive comment

TITLE. I SUGGEST: Prevalence of stillbirth and its associated factors among women giving birth in healthcare facilities in southern Ethiopia

No dot in the title please

Response: thank you for your recommendation; we have corrected and highlighted it as your suggestion. 

OBJECTIVE. I SUGGEST: To assess the prevalence and associated factors of stillbirth among women giving birth at public hospitals of Wolaita zone, southern Ethiopia

Response: thank you for your comment; we have corrected it as your suggestion.

RESULTS: I suggest you work with the language to make it less technical. Please replace labor with birth please. A woman give birth …“This finding showed that 8.7% of [95% CI: 6.5-10.8] the stillbirth was among women giving birth in public hospitals. Being rural residence [Adjusted OR = 2.57, 95% CI: 1.23, 5.40], number of pregnancies [Adjusted OR = 3.82, 95% CI: 1.17-12.47], complication during pregnancy and birth [Adjusted OR = 6.23, 95% CI: 2.67-14.58], history of stillbirth [Adjusted OR = 6.89, 95% CI: 2.57-13.57], type of birth [Adjusted OR = 7.13, 95% CI: 2.71-18.73] were found to be factors associated with stillbirth.”

Please choose to call them woman/women and not mothers and please do not mix from paragraph to paragraph: “The prevalence of stillbirth among mothers who gave birth i….

FORMAT: Make sure you do have space between the words: “737 mothers.A pre-tested” The format is still not fixed. It need to look like it should in the journal.

You need to re-work it again. Please take help from someone who can support you to make it look like a manuscript for submission to PLOS ONE. You have a good message to tell but you are jet not there. Please improve more.

Response: thank you for your comment; we have improved it as much as possible.

Reviewer #3: The authors have attended to a number of the comments raised in the first submission. However, a number of issues remain unaddressed. Generally, the manuscript still require editing as a number of statements are not very clear. For example, under Data collection process, the statement "A pretest was conducted on 5% (37 participants) of the sample size is not selected hospitals in the Wolaita zone and every possible correction was taken" This is not clear and could be completely removed as a similar statement is repeated in the Data quality control section.

Response: thank you for your comment. We have removed it.

On the point 3 I raised the authors have responded that all predictor variables with a p-value less than 0.25 during bivariable regression were entered into a multivariable regression. This is a fair answer but the question still remains how the variables that were entered in the bivariable regressions were selected initially. Some sort of statement justifying these or conceptual framework could be better. Identification of potential confounders can not be left to statistical models entirely.

Response: thank you for your comment. We have included it in the revised manuscript with track change.

The authors' response to point 4 is missing. A statement on missing data and any assumed missing data mechanism could help. I assume the the authors carried out complete case analyses which may be valid under certain missing data mechanism assumptions such as MCAR

Response: thank you for your meticulous observation. We have included it in the revised manuscript.

---

## [Decision Letter · Decision Letter 2]

25 Aug 2022

PONE-D-21-20973R2Prevalence of Stillbirth and associated factors among deliveries attended in Health Facilitiesin Southern EthiopiaPLOS ONE

Dear Dr. Haile,

Thank you for submitting your manuscript to PLOS ONE. After careful consideration, we feel that it has merit but does not fully meet PLOS ONE’s publication criteria as it currently stands. Therefore, we invite you to submit a revised version of the manuscript that addresses the points raised during the review process.

 Please be sure to pay particular attention to the following reviewer concerns: Please edit the abstract such that an interpretation of the findings described in the manuscript are included, to aid with readability and clarify the main manuscript messages for the reader. Please correct manuscript statements regarding data analysis and assumptions for missing data. Please carefully review the entire manuscript and correct any formatting issues. In particular, be sure to correct any remaining typos, such as missing spaces between words (e.g., no space between "Facilities" and "in" in the title) and between sentences, ensure that font size is consistent throughout, ensure that text is double-spaced, etc. As noted below, PLOS ONE does not copyedit accepted manuscripts, so the language in submitted articles must be clear, correct and unambiguous. Any typographical or grammatical errors should be corrected at revision. Address all remaining reviewer comments/concerns. Please submit your revised manuscript by Oct 09 2022 11:59PM. If you will need more time than this to complete your revisions, please reply to this message or contact the journal office at plosone@plos.org. Please include the following items when submitting your revised manuscript:A rebuttal letter that responds to each point raised by the academic editor and reviewer(s). You should upload this letter as a separate file labeled 'Response to Reviewers'.A marked-up copy of your manuscript that highlights changes made to the original version. You should upload this as a separate file labeled 'Revised Manuscript with Track Changes'.An unmarked version of your revised paper without tracked changes. You should upload this as a separate file labeled 'Manuscript'.If applicable, we recommend that you deposit your laboratory protocols in protocols.io to enhance the reproducibility of your results. Protocols.io assigns your protocol its own identifier (DOI) so that it can be cited independently in the future. For instructions see: https://journals.plos.org/plosone/s/submission-guidelines#loc-laboratory-protocols. Additionally, PLOS ONE offers an option for publishing peer-reviewed Lab Protocol articles, which describe protocols hosted on protocols.io. Read more information on sharing protocols at https://plos.org/protocols?utm_medium=editorial-email&utm_source=authorletters&utm_campaign=protocols.

We look forward to receiving your revised manuscript.

Kind regards,

Jennifer Manuzak

Academic Editor

PLOS ONE

Journal Requirements:

Reviewers' comments:

Reviewer's Responses to Questions

**Comments to the Author**

1. If the authors have adequately addressed your comments raised in a previous round of review and you feel that this manuscript is now acceptable for publication, you may indicate that here to bypass the “Comments to the Author” section, enter your conflict of interest statement in the “Confidential to Editor” section, and submit your "Accept" recommendation.

Reviewer #2: (No Response)

Reviewer #3: (No Response)

2. Is the manuscript technically sound, and do the data support the conclusions?

Reviewer #2: Partly

Reviewer #3: Yes

3. Has the statistical analysis been performed appropriately and rigorously? 

Reviewer #2: Yes

Reviewer #3: Yes

4. Have the authors made all data underlying the findings in their manuscript fully available?

Reviewer #2: Yes

Reviewer #3: Yes

5. Is the manuscript presented in an intelligible fashion and written in standard English?

Reviewer #2: Yes

Reviewer #3: No

6. Review Comments to the Author

Reviewer #2: Prevalence of Stillbirth and associated factors among deliveries attended in Health

Facilitating Southern Ethiopia has a valuable message that I wish could be published. Still you need to work on readability and format. Revisit the instructions for authors please and make someone support you in editing the manuscript in terms of format for instance it should be the same format everywhere. In the abstract you could also help the reader to understand the important message rather than just parroting a bunch of statistical data. You know you have to help the reader to interpret your results please.

Good luck with the third revision and please improve.

Reviewer #3: The authors have addressed the issues raised in the previous review but unfortunately the manuscript is still not suitable for publication in its current state. The general formatting of the manuscript requires; spaces between words, spaces after a fulstop etc

The statement by the authors stating "Missing data was managed by using MCAR" seem to suggest that MCAR is a technique for handling missing data which is not quite correct. I think stating that analyses were conducted assuming data was missing completely at random (MCAR) would be better.

7. PLOS authors have the option to publish the peer review history of their article (what does this mean?). If published, this will include your full peer review and any attached files.

Reviewer #2: No

Reviewer #3: No

---

## [Author Response · Author response to Decision Letter 2]

27 Sep 2022

Review Comments to the Author

Reviewer #2: Prevalence of Stillbirth and associated factors among deliveries attended in Health

Facilitating Southern Ethiopia has a valuable message that I wish could be published. Still you need to work on readability and format. Revisit the instructions for authors please and make someone support you in editing the manuscript in terms of format for instance it should be the same format everywhere. In the abstract you could also help the reader to understand the important message rather than just parroting a bunch of statistical data. You know you have to help the reader to interpret your results please.

Good luck with the third revision and please improve.

Response: thank you for your comment; we have corrected and highlighted in the revised manuscript. 

Reviewer #3: The authors have addressed the issues raised in the previous review but unfortunately the manuscript is still not suitable for publication in its current state. The general formatting of the manuscript requires; spaces between words, spaces after a fulstop etc

The statement by the authors stating "Missing data was managed by using MCAR" seem to suggest that MCAR is a technique for handling missing data which is not quite correct. I think stating that analyses were conducted assuming data was missing completely at random (MCAR) would be better.

Response: thank you for your valuable comment; we have corrected and highlighted in the revised manuscript.

---

## [Editor Report · Decision Letter 3]

4 Oct 2022

Prevalence of Stillbirth and associated factors among deliveries attended in Health Facilities in Southern Ethiopia

PONE-D-21-20973R3

Dear Dr. Haile,

We’re pleased to inform you that your manuscript has been judged scientifically suitable for publication and will be formally accepted for publication once it meets all outstanding technical requirements.

Kind regards,

Jennifer Manuzak

Academic Editor

PLOS ONE
---

## [Editor Report · Acceptance letter]

2 Dec 2022

PONE-D-21-20973R3 

Prevalence of Stillbirth and associated factors among deliveries attended in Health Facilities in Southern Ethiopia 

Dear Dr. Haile:

I'm pleased to inform you that your manuscript has been deemed suitable for publication in PLOS ONE. Congratulations! Your manuscript is now with our production department. 

Kind regards, 

on behalf of

Dr. Jennifer Manuzak 

Academic Editor

PLOS ONE